# Effect of cx-DHED on Abnormal Glucose Transporter Expression Induced by AD Pathologies in the 5xFAD Mouse Model

**DOI:** 10.3390/ijms231810602

**Published:** 2022-09-13

**Authors:** Jinho Kim, ShinWoo Kang, Keun-A Chang

**Affiliations:** 1Gachon Advanced Institute for Health Science and Technology, Graduate School, Gachon University, Incheon 21999, Korea; 2Neuroscience Research Institute, Gachon University, Incheon 21565, Korea; 3Department of Pharmacology, College of Medicine, Gachon University, Incheon 21999, Korea; 4Department of Molecular Pharmacology and Experimental Therapeutics, Mayo Clinic, Rochester, MN 55905, USA

**Keywords:** cx-DHED, Alzheimer’s disease, glucose transport, GSK-3β, O-GlcNac

## Abstract

Alzheimer’s disease (AD) is a form of dementia associated with abnormal glucose metabolism resulting from amyloid-beta (Aβ) plaques and intracellular neurofibrillary tau protein tangles. In a previous study, we confirmed that carboxy-dehydroevodiamine∙HCl (cx-DHED), a derivative of DHED, was effective at improving cognitive impairment and reducing phosphorylated tau levels and synaptic loss in an AD mouse model. However, the specific mechanism of action of cx-DHED is unclear. In this study, we investigated how the cx-DHED attenuates AD pathologies in the 5xFAD mouse model, focusing particularly on abnormal glucose metabolism. We analyzed behavioral changes and AD pathologies in mice after intraperitoneal injection of cx-DHED for 2 months. As expected, cx-DHED reversed memory impairment and reduced Aβ plaques and astrocyte overexpression in the brains of 5xFAD mice. Interestingly, cx-DHED reversed the abnormal expression of glucose transporters in the brains of 5xFAD mice. In addition, otherwise low O-GlcNac levels increased, and the overactivity of phosphorylated GSK-3β decreased in the brains of cx-DHED-treated 5xFAD mice. Finally, the reduction in synaptic proteins was found to also improve by treatment with cx-DHED. Therefore, we specifically demonstrated the protective effects of cx-DHED against AD pathologies and suggest that cx-DHED may be a potential therapeutic drug for AD.

## 1. Introduction

Alzheimer’s disease (AD), the most common form of dementia in later life, is characterized by amyloid-beta (Aβ) plaques and intracellular neurofibrillary tangles resulting in synaptic loss [1,2,3,4]. According to the previous report, the number of patients with dementia is estimated to reach 131.5 million by 2050, and as such, several preclinical or clinical studies have attempted to develop drugs for AD [5,6]. Although several different types of drugs for AD, such as pharmacologic compounds or monoclonal antibodies, have been continuously developed and expanded in clinical trials, these were complicated by toxicities or unanticipated side-effects, and, as such, there is still no definite prevention or medicine [7].

Dehydroevodiamine∙HCl (DHED) is an active component purified from the plant *Evodia rutaecarpa Bentham* [8]. According to previous studies, DHED, an acetylcholinesterase inhibitor, improves cognitive impairment, learning and memory dysfunction, as well as neuronal loss [9]. DHED also has diverse biological activities, including anti-inflammatory, anti-oxidant, and anti-tumor properties [10,11,12]. In addition, DHED has been shown to attenuate calyculin A-induced tau hyperphosphorylation in rat brain slices and oxidative stress damage in an Aβ-infused rat model [13,14]. In our previous study, we confirmed that carboxy-DHED (cx-DHED), a derivative of DHED that is highly soluble in water, improved cognitive dysfunction and reduced hyperphosphorylated tau in 5xFAD mice, an AD model presenting with Aβ plaques and hyperphosphorylated tau protein as well as abnormal glucose metabolism [15,16].

AD is a metabolic disorder related to reduced cerebral glucose metabolism, caused by both the Aβ and tau protein abnormalities [17,18]. This abnormal glucose mechanism has been observed in the brains of both patients and animal models with AD in previous studies [16,19]. For example, several studies have varyingly shown that glucose metabolism dysfunction increased the risk of cognitive impairment, while human patients with AD showed reduced glucose metabolism in the brain [20,21,22]. In contrast, increasing glucose uptake in the brain improved Aβ-induced toxicity in a fly mutant model [23,24]. Cerebral glucose metabolism is associated with glucose transportation and various glucose transporters are in the brain [21]. Glucose transporters 1 and 3 (GLUT1 and GLUT3, respectively) are considered to play a key role in regulating brain glucose transport [25]. Glucose enters the brain via the GLUT1 transporter located on blood–brain barrier (BBB) endothelial cells and astrocytes [25]. GLUT3 is predominantly distributed in neurons where it facilitates the continuous supply of glucose [25]. In a clinical study, the concentrations of GLUT1 and GLUT3 were found to be decreased in the cerebral cortex of patients with AD [26]. Although the level of glucose transporters 2 (GLUT2) has been poorly studied, one study showed that its expression level showed dynamic changes in the brain of humans with AD [27,28].

O-GlcNAcylation is a post-translational modification that involves the attachment of serine and threonine to cytoplasmic and nuclear proteins by the enzyme O-linked β-N-acetylglucosamine (O-GlcNac) [29,30]. O-GlcNAcylation is regulated by only two enzymes: O-GlcNac transferase, which attaches O-GlcNac to target proteins, and O-GlcNAcase, which cleaves O-GlcNac from target proteins [31,32,33]. Several studies have suggested that attenuated O-GlcNAcylation is associated with increased AD pathology [34,35]. For example, decreased O-GlcNAcylation was correlated with decreased levels of GLUT1 and GLUT3, which was shown to be negatively correlated with the levels of phosphorylated tau in the brains of AD patients [27]. Conversely, the increase in O-GlcNac by pharmacological treatment such as Triamet-G, an O-GlcNAcase inhibitor, prevented cognitive impairment in the brain of an AD model [34,36]. In addition, impaired insulin signaling in the brains of streptozotocin-treated rats, a model of sporadic AD, resulted in the overactivation of GSK-3β, decreased O-GlcNAcylation, tau protein hyperphosphorylation, and neurofibrillary degeneration [37]. Therefore, enhancing O-GlcNAcylation may have therapeutic effects in patients with AD.

Abnormal glucose metabolism is an important risk factor for AD, and cx-DHED may affect the treatment of AD by restoring glucose metabolism. In this study, we aimed to observe the effects of cx-DHED on the restoration of abnormal glucose transporters, O-GlcNAcylation, and memory impairment in a 5xFAD mouse model.

## 2. Results

### 2.1. cx-DHED Reversed Cognitive Impairment in 5xFAD Mice

To confirm the effect of cx-DHED on AD-related memory decline, 4-month-old WT and 5xFAD mice were treated with cx-DHED or vehicle for 2 months. After treatment with cx-DHED, we performed behavioral testing to confirm the memory changes induced by cx-DHED, and further sacrificed the mice to analyze the pathologies in the brains of vehicle-treated wild-type (WT-V), vehicle-treated 5xFAD (5xFAD-V), and cx-DHED-treated 5xFAD (5xFAD-cx-DHED) mice (Figure 1A). For the behavioral testing, novel object recognition (NOR), Y-maze, and a passive avoidance test were performed sequentially. The exploration ratio in the NOR of the WT-V (76.15 ± 3.36 s, *p* < 0.0001) and 5xFAD-cx-DHED (80.04 ± 4.959 s, *p* < 0.0001) groups was significantly increased compared to the exploration ratio of the familiar object (WT-V, 23.85 ± 3.36 s; 5xFAD-cx-DHED, 19.96 ± 4.96 s) (Figure 1B); however, the 5xFAD-V group stayed longer in the familiar object area (69.325 ± 11.43 s, *p* < 0.01) than in the novel object area (30.68 ± 11.43 s). Locomotor activity did not differ between groups during the test. In the Y-maze test, spontaneous alteration was significantly decreased in 5xFAD-V mice (52.74 ± 5.93%, *p* < 0.01) compared to that in control mice (76.67 ± 3.73%). Interestingly, cx-DHED-treated 5xFAD mice (80.37 ± 2.21%, *p* < 0.001) showed increased spontaneous alteration compared to 5xFAD-V mice (Figure 1C). In the passive avoidance test, the latency to enter the dark room was measured on the test day. The latency time in the 5xFAD-V group (91.17 ± 18.72 s, *p* < 0.0001) was significantly increased compared to that in the WT-V group (254.5 ± 6.325 s). However, treatment with cx-DHED in 5xFAD mice (239.3 ± 4.539 s, *p* < 0.0001) was reversed compared to that in the 5xFAD-V group. These results indicated that cx-DHED treatment attenuated memory deficits in mice with AD.

### 2.2. cx-DHED Suppressed the Formation of Amyloid Plaque

To analyze the effects of cx-DHED on amyloid pathologies, we stained the brains of mice with AD using thioflavin S. We found that the accumulation of amyloid plaques was decreased in both the hippocampus and cortex of 5xFAD-cx-DHED mice compared with 5xFAD-V mice (Figure 2A). In the hippocampus, the number of plaques in the 5xFAD-cx-DHED group (14.4 ± 2.337, *p* < 0.05) was significantly lower than in the 5xFAD-V group (26.8 ± 3.338, Figure 2B). Furthermore, the plaque burden in the cortex of the 5xFAD-cx-DHED group (66.4 ± 7.467, *p* < 0.05) was significantly lower than that in the cortex of the 5xFAD-V group (90.8 ± 6.981, Figure 2C). We found that cx-DHED suppressed amyloid plaque formation.

### 2.3. cx-DHED Reduced Astrogliosis in 5xFAD Mice Brains

Astrogliosis is defined as astrocytes undergoing morphological, molecular, and functional remodeling in response to disease and is a characteristic of AD [38]. We observed a change in astrogliosis in the mouse brain following cx-DHED treatment. In immunofluorescence staining using the GFAP antibody (Figure 3A), we found that astrocyte expression in the cortex of the 5xFAD-V group (9.96 ± 0.2591) was significantly higher than that in the WT-V group (1 ± 0.4244, *p* < 0.0001). Treatment of 5xFAD mice with cx-DHED (5.727 ± 0.392, *p* < 0.0001) significantly reduced the expression of GFAP in the 5xFAD compared to the 5xFAD-V group (Figure 3B). We further confirmed the overexpression of GFAP in the cortex of the 5xFAD-V group (2.577 ± 0.154, *p* < 0.0001) compared to WT-V mice (1 ± 0.122, Figure 3C). However, cx-DHED treatment significantly restored the overexpression levels of GFAP in 5xFAD mice (2.063 ± 0.049, *p* < 0.05) compared to those observed in 5xFAD-V mice.

### 2.4. cx-DHED Reversed the Abnormal Expression of Glucose Transporter in 5xFAD Mice Brains

Abnormal glucose metabolism and expression levels of glucose transporters in the brains of human patients and mouse models of AD have been previously reported [26,39,40]. Therefore, we confirmed the protein levels of the glucose transporter in the brains of mice using Western blot (Figure 4A). The expression level of GLUT1 was significantly lower in the cortex of the 5xFAD-V group (0.246 ± 0.03, *p* < 0.01) than the WT-V group (1 ± 0.1448). Interestingly, cx-DHED treatment recovered the protein level of GLUT1 in the brains of 5xFAD mice (0.717 ± 0.144, *p* < 0.05) compared to 5xFAD-V mice (Figure 4B). In contrast, the protein level of GLUT2 was significantly higher in the cortex of the 5xFAD-V group (2.179 ± 0.184, *p* < 0.0001) than in that of the WT-V group (1 ± 0.11). However, the GLUT2 protein level in the cortex of 5xFAD-cx-DHED mice (1.213 ± 0.117, *p* < 0.001) was restored to the level in the WT-V group (Figure 4C). Regarding GLUT3 expression level in the cortex, levels in the 5xFAD-V group (0.662 ± 0.079, *p* < 0.05) were significantly lower than those in the WT-V group (1 ± 0.033). Following treatment with cx-DHED, the protein level of GLUT3 in 5xFAD mice (1.011 ± 0.112, *p* < 0.05) returned to the levels observed in the WT-V group. Our results showed that treatment with cx-DHED exerted a therapeutic effect on the abnormal expression of glucose transporters in the brains of an AD mouse model.

### 2.5. cx-DHED Increased the Levels of O-GlcNAcylation but Decreased Phosphorylation of GSK-3β in 5xFAD Mice Brains

Impairment of glucose utilization leads to a decrease in O-GlcNAc, resulting in aggravated amyloid and tau pathologies in the brains of AD patients [41]. Therefore, we examined the effect of cx-DHED on O-GlcNAcylation in the cortex of 5xFAD mice. In the brains of the 5xFAD-V group (0.630 ± 0.035, *p* < 0.01), the level of O-GlcNac was significantly lower than that in the brains of the WT-V group (1 ± 0.06, Figure 5A). Interestingly, treatment with cx-DHED significantly increased the level of O-GlcNac in 5xFAD mouse brains (0.785 ± 0.022, *p* < 0.05).

In a prior study, we identified that cx-DHED treatment attenuates hyperphosphorylated tau in 5xFAD mouse brains [15]. In this study, we found that GSK-3β phosphorylation is similarly associated with tau pathology in model mice. The level of tyrosine 216 phosphorylation of GSK-3β was significantly higher in the 5xFAD-V group (1.334 ± 0.024, *p* < 0.0001) than in the WT-V group (1 ± 0.047, Figure 5B). However, treatment with cx-DHED significantly reduced the phosphorylation of GSK-3β in the brains of 5xFAD-V mice (0.9191 ± 0.025, *p* < 0.0001). The level of total GSK-3β in the mouse brain did not significantly differ between groups. These results showed that cx-DHED treatment attenuated AD pathologies in 5xFAD mice by enhancing O-GlcNAcylation and suppressing the phosphorylation of GSK-3β.

### 2.6. cx-DHED Treatment Attenuated Loss of the Synaptic Protein in 5xFAD Mice Brains

Synaptic loss is strongly correlated with dementia severity [42]. Therefore, we observed alterations in synaptic proteins in the brains of mice after cx-DHED treatment. In the synaptic fraction of the cortex sample, we found that the expression level of PSD-95 in the 5xFAD-V group (0.638 ± 0.055, *p* < 0.05) was significantly lower than that in the WT-V group (1 ± 0.051, Figure 6A); however, cx-DHED treatment in 5xFAD mice (0.895 ± 0.08, *p* < 0.05) significantly increased the expression of PSD-95. In addition, the expression levels of synaptophysin, a presynaptic marker, was significantly decreased in the brains of 5xFAD mice (0.736 ± 0.038, *p* < 0.01) compared to those in WT-V mice (1 ± 0.063, Figure 6B). This reduction was reversed by cx-DHED treatment in 5xFAD mice (0.891 ± 0.026, *p* < 0.05). These results suggest that cx-DHED may reverse AD pathologies, including glucose metabolism dysfunction, to ultimately enhance synaptic stability and cognitive function.

## 3. Discussion

In our study, we observed the effect of cx-DHED on AD pathologies associated with abnormal glucose metabolism. Briefly, we treated male 5xFAD model mice with cx-DHED for 2 months and performed behavioral tests for cognitive function. cx-DHED restored memory impairment, reduced amyloid plaques, and restored astrogliosis in the brain. Furthermore, cx-DHED restored the abnormal expression levels of GLUTs, enhanced O-GlcNAcylation, and reduced the phosphorylation of GSK-3β in 5xFAD mice. As a result, cx-DHED treatment was found to prevent the loss of synaptic proteins in 5xFAD mice.

In our previous study, we confirmed that cx-DHED was delivered to the brain after penetration of the BBB using LC–MS [15]. Furthermore, cx-DHED has the advantage of high bioavailability thanks to its higher solubility in water than DHED. cx-DHED has further been shown to reduce Aβ and phosphorylated tau in 5xFAD mice [15].

AD is considered a metabolic disorder, with reduced glucose metabolism and insulin resistance in the brain strongly linked to its pathology [18]. Both clinical and preclinical studies have demonstrated that disrupted glucose metabolism is an important feature of the AD brain [39,43,44]. For example, type 2 diabetes mellitus, which is a disease associated with glucose metabolism, is related to AD, and insulin resistance and deficiency can interact with Aβ and phosphorylated tau, leading to AD [45]. Because of the hypometabolism in the AD brain, ^18^F-Fluorodeoxyglucose (FDG) PET was performed to diagnose AD progression, resulting in abnormal cerebral glucose; these results were correlated with the decline in cognitive performance [24,46]. In animal studies, FDG uptake was significantly decreased in the brains of 5xFAD mice compared to that in the brains of wild-type mice [47,48]. This metabolic change is associated with lower amounts of glucose transporters and insulin signaling in AD [49,50]. In particular, the disrupted density of GLUTs exaggerates AD pathologies in animal models [24,40]. GLUT1 and GLUT3, which are in the BBB and astrocytes, are major glucose transporters that play important roles in glucose uptake into neurons [51,52]. In the brains of patients with AD, the density of GLUT1 and GLUT3 is reduced compared to that in the normal brain, and it is considered an important cause of the dysfunction of glucose metabolism [26]. In our study, we confirmed that GLUT1 and GLUT3 densities were significantly decreased in the brains of 5xFAD mice (Figure 4). In contrast to GLUT1 and GLUT3, GLUT2 was significantly increased in the cortex of 5xFAD mice compared to the level in wild-type mice. This result may be related to astrocyte overexpression, and is consistent with the results of studies in the brains of human AD patients [53].

Aβ toxicity and metabolic dysregulation are known to decrease O-GlcNAcylation levels [41]. In addition, modulation of O-GlcNAcylation, which targets serine/threonine residues, alters neuronal and synaptic functions, and has a protective effect against disease [54]. In our study, the O-GlcNac expression level was decreased in the 5xFAD-vehicle group compared to that in the wild-type vehicle group, but cx-DHED improved O-GlcNAcylation in the brains of 5xFAD mice (Figure 5).

We further identified increased levels of phosphorylated GSK-3β (Tyr216) in 5xFAD mice, which were reversed by cx-DHED treatment. The O-GlcNAcylation and phosphorylation of GSK-3β (Tyr216) were considered to act competitively. As cx-DHED treatment enhanced O-GlcNAcylation by restoring abnormal glucose metabolism, phosphorylation of GSK-3β, phosphorylated tau, and disrupted synaptic proteins were improved in AD model mice (Figure 7). Clinical studies have demonstrated that O-GlcNAcylation regulates phosphorylated tau and is correlated with cognitive impairment in AD patients [53,55]. Therefore, cx-DHED treatment may improve glucose metabolic dysfunction, including GLUT density, O-GlcNAcylation, and memory impairment, in 5xFAD mice.

## 4. Materials and Methods

### 4.1. Animals

In this study, 5xFAD transgenic mice (The Jackson Laboratory, Bar Harbor, ME, USA) were used, with wild-type litter-mate mice used as controls. The 5xFAD mice express APP (Swedish (K670N/M671L), Florida (I716V), and London (V717I) mutations) and PS1 (M146L/L286V mutations) transgenes. Male wild-type and hemizygous 5xFAD mice were born from female F1 hybrid mice (B6SJL) obtained from JAX Laboratories. Genotyping was performed by polymerase chain reaction using DNA isolated from in-ear biopsy. The experimental groups were as follows: vehicle-treated WT mice (WT-V, n = 8); vehicle-treated 5xFAD mice (5xFAD-V, n = 6); cx-DHED-treated 5xFAD mice (5xFAD-cx-DHED, n = 7). The temperature and humidity of the breeding room were automatically maintained at 22 ± 2 °C and 50 ± 10%, respectively. A 12L:12D photoperiod was provided, and food and water were provided *ad libitum* in cages during the acclimation period. All animal experiments were approved by the Laboratory Animal Care Committee of CACU, Gachon University (LCDI-2020-0012).

### 4.2. cx-DHED Treatment

We injected 4-month-old 5xFAD or wild-type (WT) mice with 1 mg/kg of cx-DHED by intraperitoneal injection (i.p) daily for 2 months, as referred to in our previous study [15]. A 0.9% saline solution was used as the vehicle.

### 4.3. Behavior Test

After cx-DHED treatment, we performed three behavioral tests to evaluate changes in memory and cognition in mice in all groups (WT-V, n = 8; 5xFAD-V, n = 6; 5xFAD-cx-DHED, n = 7). Mice were rested for one day after each test. All tests were automatically recorded and counted using the Ethovision XT 9 system (Noldus Information Technology, Wageningen, The Netherlands). All the daily procedures were performed between 9:00 and 15:00.

### 4.3.1. Y-Maze

The Y-maze test was performed using the white polyvinyl plastic maze composed of three branches 40 cm in length, 6.8 cm in width, and 15.5 cm in height. Briefly, the mice were placed freely in the maze for 8 min. We counted the number of entries, defined as the mouse completely entering into the branch. The consecutive number of entries in the three arms was given 1 point. Spontaneous alternation was calculated by the following equation: sequence of entries/(Total entries − 2) × 100.

#### 4.3.2. Novel Object Recognition Test

For the novel object recognition test, the mice were habituated to the open-field chamber (38 cm wide × 38 cm high × 40 cm long) for 20 min on the first day. The following day, the mice were placed in the same open-field box and exposed to two identical objects for 5 min. The objects (height, 10 ± 2 cm) were colored and weighted to be heavy enough not to be displaced by the animals. The objects were positioned 5 cm away from the walls of the box in opposite corners. The mice freely explored the objects in the open-field box for 5 min and then returned to their home cage. After 24 h, one object was replaced with a novel object and we measured the exploration time of mice in familiar areas and with the novel object over 5 min. Boxes were cleaned with 70% ethanol and mice were returned to the home cage. The exploration time of each object, the velocity, and the total distance were automatically recorded using the Ethovision XT 9 system. The memory index was calculated as the difference in time as a percentage of the total time exploring the two objects.

#### 4.3.3. Passive Avoidance Test

The passive avoidance test was performed for three continuous days using a passive avoidance apparatus (Gemini Passive Avoidance System; San Diego Instruments, San Diego, CA, USA) (42.5 cm wide and 35.5 cm long), composed of two adjacent bright and dark chambers connected by a remote operational gate. The bright chamber was illuminated with a 6 W LED light. On the first day, we allowed the mice to freely explore the chambers. On the next day, when the mice entered the dark chamber, they were exposed to an electric shock to the feet (2 mA for 2 s) with the gate closed. After 24 h, we measured the latency times of the mice from their placement in the bright chamber until their entrance to the dark chamber.

### 4.4. Tissue Preparation

The mice were anesthetized with a mixture of Zoletil (8.3 mg/kg) and Rompun (15 mg/kg), and then the brains were extracted. The cortex and hippocampus were dissected from the hemisphere and were immediately frozen in liquid nitrogen for Western blot. The other hemisphere of each mouse brain was fixed in 4% paraformaldehyde at 4 °C for 24 h and then was dehydrated in a 30% sucrose solution for 3 days. The dehydrated tissues were frozen in molds filled with optimal-cutting-temperature compounds (Sakura, Osaka, Japan). After frozen tissues were cut at a thickness of 22 μm using the cryomicrotome (Cryotome, Thermo Electron Corporation, Waltham, MA, USA), they were stored in a cryoprotectant solution (ethylene 30% and glycerol 30% in PBS) at 4 °C.

### 4.5. Immunohistochemistry

To confirm astrocyte expression levels, immunofluorescence analysis was performed as previously described [56]. Briefly, after washing in PBS-T (0.2% Triton X-100 in PBS), frozen brain sections were blocked in blocking solution (0.5% BSA and 3% normal goat serum in 0.4% PBS-T) at room temperature for 1 h and then were incubated with primary antibody overnight at 4 °C in PBS-T solution (GFAP, 1:500, DAKO, Centennial, CO, USA). The following day, the tissue was incubated with Alexa Fluor 488 antibody (Invitrogen, Carlsbad, CA, USA) as a fluorescent secondary antibody for 1 h at room temperature. To stain Aβ plaques, Thioflavin S was applied to the sections for 10 min at room temperature. DAPI was used as a counterstain. Images were taken using a Nikon TS2-S-SM microscope (Nikon Microscopy, Tokyo, Japan), equipped with a Nikon DS-Qi2 camera. Once the regions of interest (ROIs) were defined, the signal was used to measure the fluorescence intensity and was converted to a percentage (n = 3–6 per group).

### 4.6. Western Blot

The brain tissues were lysed with radioimmunoprecipitation assay (RIPA) buffer (150 mM NaCl, 1% NP-40, 0.5% sodium deoxycholate, 0.1% SDS, 50 mM Tris, pH 8.0) containing protease inhibitors (Roche Applied Science, Mannheim, Germany) and a cocktail of phosphatase inhibitors (Sigma-Aldrich, St. Louis, MO, USA) on ice for 30 min. After centrifugation at 13,000 rpm for 20 min at 4 °C, the lysate was quantified using the Bradford assay (Bio-Rad Laboratories, Inc., Hercules, CA, USA), and run onto an 8 or 15% sodium dodecyl sulfate-polyacrylamide gel electrophoresis (SDS-PAGE) gel. Proteins were then transferred onto a polyvinylidene difluoride (PVDF) membrane (Merck, Kenilworth, NJ, USA). After blocking the transferred membrane with blocking buffer (5% skim milk or 3% BSA in TBS-T) at room temperature for 1 h, the membrane was incubated with the appropriate primary antibody (GFAP; DAKO, Z0334, GLUT1; Santa Cruz, Dallas, TX, USA, sc-377228, GLUT2; sc-518022, GLUT3; sc-74497, O-GlcNac, Waltham, MA, USA, Tyr216 GSK-3; sc-135653, GSK3; sc-9166, PSD-95; Thermo scientific, Waltham, MA, USA, MA1-046, Synaptophysin; abcam, Cambridge, UK, ab8049) overnight at 4 °C. After washing three times in TBS-T, the membrane was incubated in secondary antibody for 1 h at room temperature. Protein bands were detected using Enhanced Peroxidase Detection (PicoEPD) enhanced Chemiluminescent (ECL) (ELPISBIO, Daejeon, Korea) or Immobilon Western Chemiluminescent HRP Substrate (Millipore, Burlington, MA, USA) and BLUE X-ray film (AGFA, Mortsel, Belgium). Quantification of the bands was performed using the Image J software v1.4.3.67 (n = 4–6 per group).

### 4.7. Statistical Analysis

All the data are presented as the mean ± standard error of the mean (SEM). Statistical analysis was performed using GraphPad Prism 9.1.0 (221) software (GraphPad Software Inc., San Diego, CA, USA), and outliers were removed using the Outlier calculator (significance level: Alpha = 0.05) in GraphPad Prism software. All values are expressed as the mean ± standard error of the mean. Normality tests were performed using Shapiro–Wilk test. Differences in the collected data between groups were analyzed using a one-way analysis of variance (ANOVA) followed by Tukey’s multiple comparisons test or Student’s *t*-test. Statistical significance was set at *p* < 0.05.

## Figures and Tables

**Figure 1 ijms-23-10602-f001:**
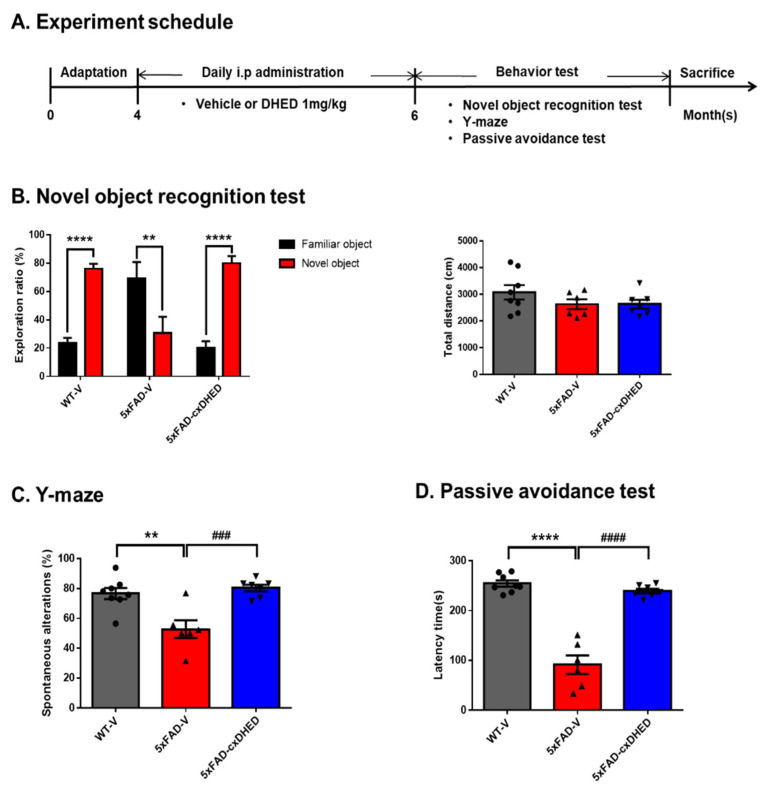
cx-DHED effects on cognitive impairment in 5xFAD mice. (**A**) The scheme showing the experimental procedure. After treatment with cx-DHED by intraperitoneal injection (1 mg/kg) for 2 months, behavioral testing was performed to determine the effect of cognitive behavior in 5xFAD mice. The behavior test was sequentially performed as follows: (**B**) novel object recognition test, (**C**) Y-maze test, and (**D**) passive avoidance test. Values are expressed as the mean ± SEM. ** *p* < 0.01, **** *p* < 0.0001 vs. WT-V, ### *p* < 0.001, #### *p* < 0.0001 vs. 5xFAD-V. Statistical analysis between the three groups was performed using one-way ANOVA, followed by Turkey’s post hoc test. Vehicle-treated wild-type (WT-V; n = 8), vehicle-treated 5xFAD (5xFAD-V; n = 6), and cx-DHED-treated 5xFAD (5xFAD-cx-DHED; n = 7) mice.

**Figure 2 ijms-23-10602-f002:**
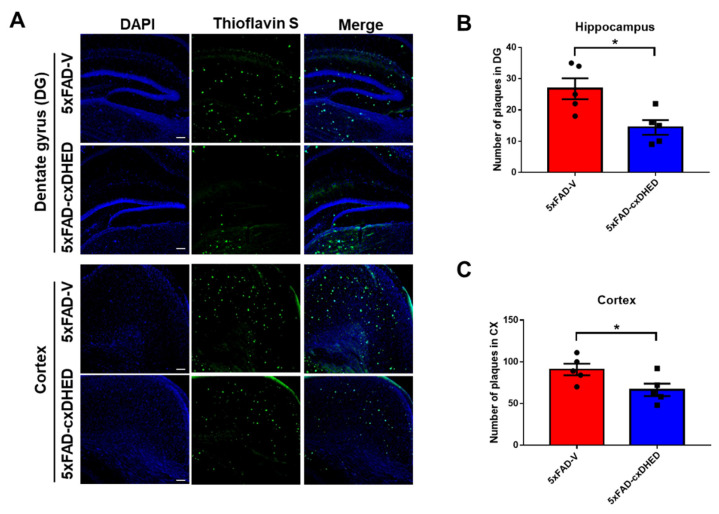
cx-DHED suppressed the formation of amyloid plaques. (**A**) Images showing the thioflavin-S stain localized to the amyloid plaques in the hippocampus (dentate gyrus) and cortex of 5xFAD mice. Quantification was determined as the number of plaques in the hippocampus (**B**) and cortex (**C**). Values are expressed as the mean ± standard error of the mean (5xFAD; n = 5, 5xFAD-cx-DHED; n = 5). Scale bars, 100 μm. * *p* < 0.05. Statistical analysis between the two groups was performed using the Student *t*-test.

**Figure 3 ijms-23-10602-f003:**
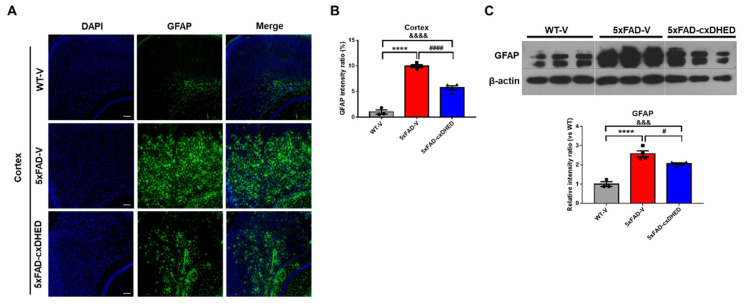
cx-DHED reduced astrogliosis in the brain of 5xFAD mice. (**A**) Representative image of GFAP staining in the cortex to detect astrogliosis and (**B**) the quantification of GFAP immunofluorescence in the cortex of mice. (**C**) Western blot was performed to confirm the astrocyte expression level in the cortex of model mice. Values are expressed as the mean ± standard error of the mean (WT-V; n = 3, 5xFAD; n = 4, 5xFAD-cx-DHED; n = 4). β-actin was used as a loading control. **** *p* < 0.0001 vs. WT-V group; &&& *p* < 0.001, &&&& *p* < 0.0001 vs. 5xFAD-cxDHED group; # *p* < 0.05 and #### *p* < 0.0001 vs. 5xFAD-V group. Scale bars, 100 μm. Statistical analysis between the three groups was performed using the one-way analysis of variance, followed by Turkey’s post hoc test.

**Figure 4 ijms-23-10602-f004:**
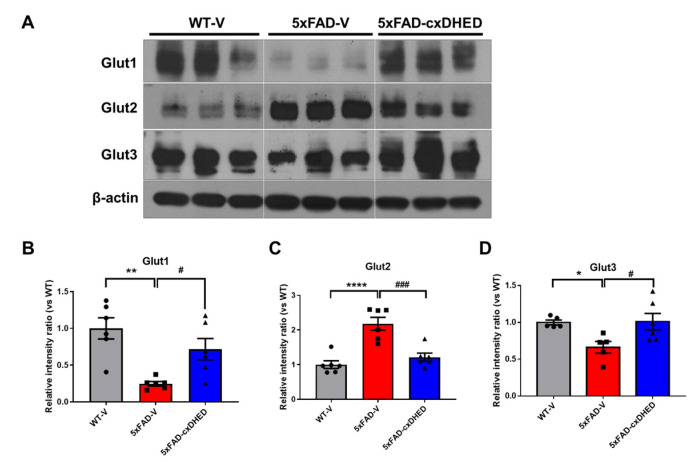
cx-DHED rescued the abnormal expressions of glucose transporters in the brain of 5xFAD mice. (**A**) Protein expression levels of glucose transporters (GLUT) in the cortex of mice. (**B**) GLUT1 (WT-V; n = 6, 5xFAD; n = 6, 5xFAD-cx-DHED; n = 6), (**C**) GLUT2 (WT-V; n = 6, 5xFAD; n = 6, 5xFAD-cx-DHED; n = 6), and (**D**) GLUT3 (WT-V; n = 5, 5xFAD; n = 5, 5xFAD-cx-DHED; n = 6) expression levels were normalized to β-actin. Values are expressed as the mean ± standard error of the mean. * *p* < 0.05, ** *p* < 0.01, and **** *p* < 0.0001 vs. WT-V group; # *p* < 0.05 and ### *p* < 0.001 vs. the 5xFAD-V group. Statistical analysis between the three groups was performed using the one-way analysis of variance, followed by Turkey’s post hoc test.

**Figure 5 ijms-23-10602-f005:**
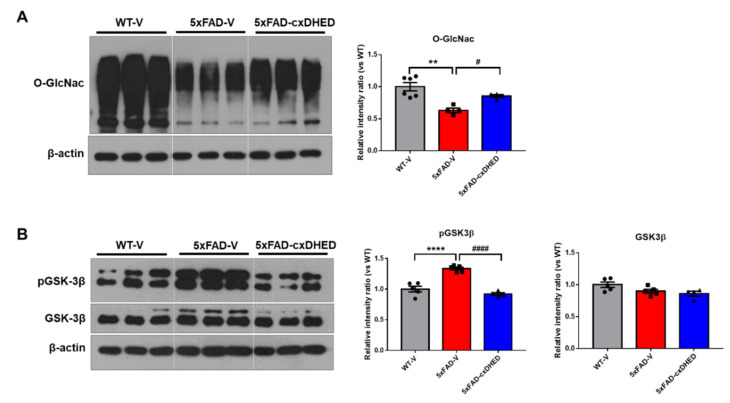
cx-DHED increased O-GlcNAcylation, but decreased phosphorylation of GSK-3β in the brain of 5xFAD mice. (**A**) Representative blot and quantification of O-GlcNac expression in the cortex of the mice (WT-V; n = 6, 5xFAD; n = 4, 5xFAD-cx-DHED; n = 4). (**B**) The representative blot and quantification of the phosphorylation of GSK-3β (pGSK-3β) and total GSK-3β (WT-V; n = 5, 5xFAD; n = 5, 5xFAD-cx-DHED; n = 4). β-actin was used as a loading control. Values are expressed as the mean ± SEM. ** *p* < 0.01 and **** *p* < 0.0001 vs. WT-V group; # *p* < 0.05 and #### *p* < 0.001 vs. 5xFAD-V group. Statistical analysis between the three groups was performed using the one-way analysis of variance, followed by Turkey’s post hoc test.

**Figure 6 ijms-23-10602-f006:**
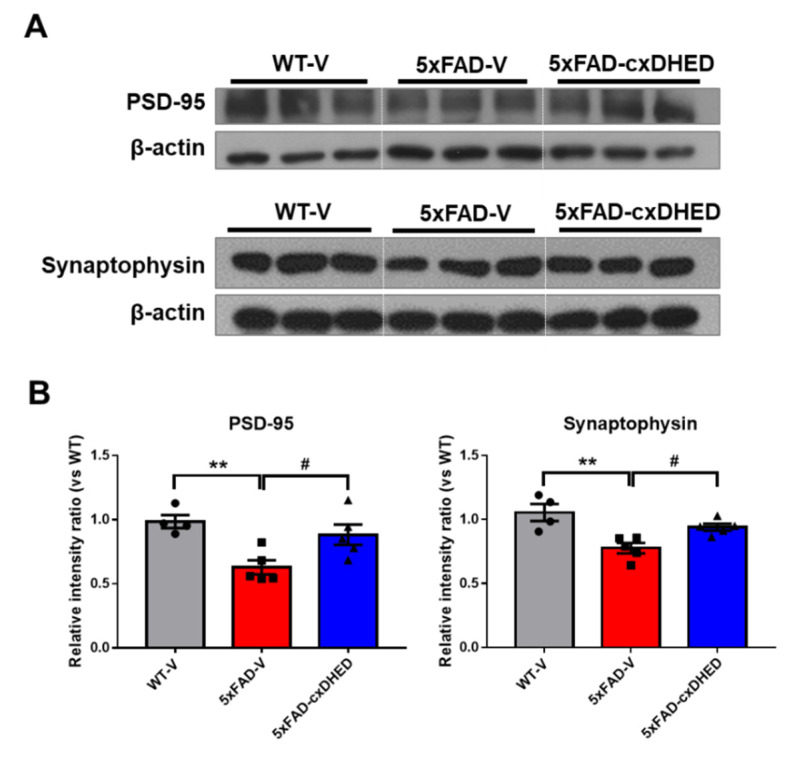
cx-DHED treatment alleviated loss of synaptic protein in the brain of 5xFAD mice. (**A**) The expression levels of PSD-95, a postsynaptic marker, and synaptophysin, presynaptic markers, were analyzed in the fraction of synaptic protein in the brain of mice. (**B**) Quantified values of PSD-95 and synaptophysin are expressed as the mean ± SEM (WT-V; n = 4, 5xFAD; n = 5, 5xFAD-cx-DHED; n = 5). β-actin was used as a loading control. ** *p* < 0.011 vs. WT-V group, # *p* < 0.05 vs. 5xFAD-V group. Statistical analysis between the three groups was performed using the one-way analysis of variance, followed by Turkey’s post hoc test.

**Figure 7 ijms-23-10602-f007:**
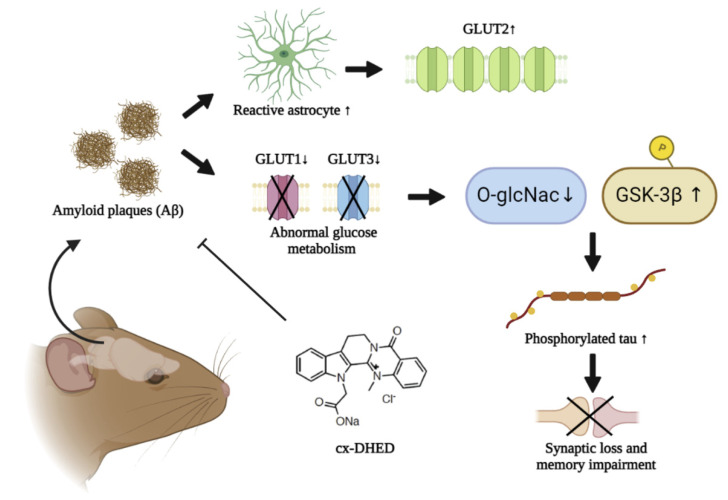
Study summary. In the brain of 5xFAD mice, accumulation of amyloid plaques (Aβ) perturbs normal glucose metabolism by driving a reduction in glucose transporters 1 and 3 (GLUT1 and GLUT3, respectively) or overexpression of GLUT2 by upregulated reactive astrocytes. While phosphorylated GSK-3β was increased, O-GlcNac was diminished by dysfunctional glucose metabolism. Consequently, increased phosphorylated tau leads to synaptic loss and memory impairment. However, cx-DHED treatment attenuated the formation of amyloid plaques and upregulated O-GlcNac, as well as the level of GLUT in the brain of 5xFAD mice. Finally, cx-DHED treatment led to the reduction in phosphorylated tau and recovery of memory function.

## Data Availability

The data supporting the findings of this study are available from the corresponding author upon reasonable request.

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
