# Peer review of "Effect of cx-DHED on Abnormal Glucose Transporter Expression Induced by AD Pathologies in the 5xFAD Mouse Model"

_ijms, 2022, doi:10.3390/ijms231810602_

Round 1
Reviewer 1 Report
The authors have demonstrated the biological effects of the DHED derivative carboxy-dehydroevodiamine∙HCl (cx-DHED). the article is very well written overall with minor english corrections as well as interesting and well written results. They have advanced their research to further the explanation and biological effect of the molecule. My only request would be that they strengthen the introduction and conclusion to compare these results with other systems that target the same pathway to demonstrate if it could be better or used together with other treatments.
The authors have demonstrated the biological effects of the DHED derivative carboxy-dehydroevodiamine∙HCl (cx-DHED). They have advanced their research from their previous article (citation 11), in which they showed the derivative cx/DHED could improve memory deficits, to elucidate the pathway in which this molecule functions (currently unknown in the literature. novelty). They did this by identifying the fact that the compound affects astrogliosis and the glucose receptors (which has already been linked to AD and therefore makes sense). There are several suggestions as follows:
Suggested Changes
Fig 1 should be better cited as even if the reviewers redid the experiments it is just to prove the same effect already shown in citation 11.
Figure3: C is missing in the figure description
Figure 4..... Glut 2 calculations seem to be off as the difference in the western of WTV is clearly much less intense compared to both 5xFad and cxDHED meaning that the compound and the wt can not be identical as shown in the graph.d
Figure 5 same. looking at the gels its impossible to say WTV is the same with 5xFAD when looking at the integration box given in the supplemental. please demonstrate what was actually used to determine intensity as the single band at the lowest point indeed would make sense
The article is very well written overall with minor English corrections needed. ex:
line 33 for ad [5,6]; however, ..... is the correct English usage of however in this case
line 43 "metabolism, and both" should be changed to " metabolism, in which both"
lines 44-48 can be combined to make a sentence more reader friendly as it all states issues with the glucose metabolism in different studies.
line 58-60 can be written more distinctly to make a more fluid statement
line 66-67 "with decreased GLUT1 and GLUT3 and 66 levels of tau phosphorylation in the brains of patients with AD" can be changed to " GLUT1, GLUT3, and....."
Line 72 may 72 treat AD by restoring abnormal glucose metabolism and managing AD pathologies" treat AD and manage AD pathologies is redundant and one of them can be removed
lines 84-85 " a novel object recognition test, Y-maze, and passive avoidance tests were performed sequentially" should be changed to " an novel recognition-, y-maze-, and passive avoidance test..." (hyphens could also be left out depending on the preference of the journal)
line 88 (Fig. 1B). However, should be changed to Fig. 1b); however,
Line 132 " We also 132 confirmed that astrogliosis significantly increased in the 5xFAD-V group (2.577 ± 0.154, 133 p<0.0001) compared to the WT-V mice (1 ± 0.122, Fig.3C)" this is written backwards... the vehicle does not increase the expression.... that is the expression in 5xfad mice.... the compound decreases it. should be written as such.
line 150 " cortex of the 5xFAD-V group (0.246 ± 0.03, p < 0.01) than in that" than in that is incorrect...... respective to that in.
line 147 " AD have been reported previously" should be changed to "have been previously reported"
line 173 " n O-GlcNAc and results in ag" change to "glcNAc resulting in"
Line 203 same as previous Fig. 6A). However, to "; however,"
line 220 " In our study, we improved the effect of cx-DHED on AD pathologies associated with 220 abnormal glucose metabolism." this is untrue as you have not IMPROVED the effect of the compound. you just furthered the understanding of its mechanism.
Major Change:
The authors should strengthen the introduction and conclusion to compare these results with other systems that target the same pathway to demonstrate if it could be better or used together with other treatments. SEveral suggested review citations include
https://doi.org/10.3390/ijms22169082
DOI: 10.1126/sciadv.abc7031
https://doi.org/10.1016/j.jconrel.2020.05.044
https://doi.org/10.2174/1381612823666170828133059
and the authors are asked to find experimental articles where other compounds have shown similar effects by affecting these pathways in 5xFAD mice.
Reviewer 2 Report
The current study attempts to improve the effect of cx-DHED on AD pathologies associated with abnormal glucose metabolism. Cx-DHED is a derivative of DHED, which is an alkaloid purified from Evodia rutaecarpa Bentham. The authors showed that cx-DHED restored memory impairment, reduced amyloid plaques, and restored astrogliosis in the brain. In addition, cx-DHED restored the abnormal expression levels of GLUTs, enhanced O-Glc-NAcylation, and reduced the phosphorylation of GSK-3β in 5xFAD mice. As a result, cx-DHED treatment prevented the loss of synaptic proteins in 5xFAD mice.
I found the paper to be overall very well written and I felt confident that the authors performed careful research analysis and data interpretation. I recommend that only a minor revision of the manuscript is warranted. I explain my concerns in more detail below and I would ask that the authors will correct them.
Minor comments:
Line 15 : Please add protein after tau, otherwise tau would be a letter of the Greek alphabet, that cannot be phosphorylated.
Line 35-36: please explain for readers who are not directly in this field what Evodia rutaecarpa Bentha is; which must be written in italics! Also, provide detailed info about (DHED) is an active component which from chemical point of view is an alkaloid with diverse biological activities including anti-inflammatory, anti-obesity, and antitumor.
Line 44: again add protein after tau, and check all the text !
Line 84: You may use the abbreviation NOR for the method (novel object recognition) to simplify reading the text.
Line 127: Please avoid very short sentence such as “Astrogliosis is a characteristic of AD” especially when we significantly work to provide evidence for this defense mechanism. Please describe shortly in 1-2 sentence what astrogliosis is ?
Line 148-149: “Therefore, we confirmed the protein levels of the glucose transporter in the brains of mice.” I consider that will be helpful to continue the sentence by adding the methods used to determine the protein content.
The text of the figures should be enlarged by at least 2-3 font sizes to be more easily visible.
Fig 3. Please move the bar graph to the upper right, next to the one and enlarge the image with WB to gain visibility. Also, explain the use of β-actine, this appears only once in the text (at line 166) but is used in several figures! In the legend of Fig. 3, A and C do not appear explicitly.
Fig 4. , Fig5., Fig 6. (A) is missing again ? even if it is understood from the legend, A must also be included.
Reviewer 3 Report
Attached

Round 2
Reviewer 3 Report
Authors have taken good efforts to answer all the queries raised in the previous review. Manuscript quality has improved with regard to experimental details however, there are several instances of grammatical errors which needs to be significantly corrected. An extensive check is recommended. In addition, there are some concerns based on the current revisions which needs to be certainly clarified for the manuscript to be considered for publication quality. These are very straight forward and authors should be able to easily incorporate them.
1) Figure 4D legend. n=5~6 needs to be corrected to the exact number used for each group (if it meant different numbers for different experiments shown as separate graphs). Similarly, figure 5B legend also needs the same correction. 'n' for animal number should always be accurate and precise for scientific experiments.
2) Previous review, question 11 raised a concern about how different 'n' value within experiments were accounted during statistical analysis. This does not seem to be answered. Please clarify.
3) Sections 4.5, 4.6, revisions in red : what is "n= 3~6 and 4~6 per animal?" Did authors mean 'per group'? Please clarify.
4) Revised lines 35-37 in introduction: 'Although several drugs for AD were continuously developed and expanded in clinical trial, there is no prevention or medicine for a variety reasons including side effect"- this statement is incomplete and vague. The introductory section is very important with regard to giving major context details. 'variety reasons including side effects'- needs significant correction.
5) Pointing out few instances which needs grammatical corrections: line 40,
lines 58-59, line 74, line 132, line 149, line 205, line 227, line 298. There are many examples. Hence, as indicated above, authors should definitely consider conducting an extensive language and grammar check and make necessary corrections.
Author Response
September 2nd, 2022
Editor-in-Chief
International Journal of Molecular Sciences
Dear Editor and Reviewers
My co-authors and I would like to thank you for allowing us to submit a revised version of our
manuscript. We are grateful to the Editor and Reviewers for their positive and constructive comments
and suggestions on how to improve our manuscript entitled “Effect of cx-DHED on abnormal glucose
transporter expression induced by AD pathologies in the 5xFAD mouse model” (ijms-1863837).
We have taken into consideration the critiques of the Reviewer and have revised our manuscript
accordingly. Please find attached point-by-point responses to the reviewer’s comments and the revised
version of our manuscript (with all changes marked in red), which we would like to submit for your
consideration.
We believe that we have significantly improved the quality of our manuscript and hope it now
reaches the standard of your esteemed journal, IJMS.
Once again, we would like to thank you and the reviewers for your insightful comments on our paper.
We look forward to hearing from you.
Sincerely,
Keun-A Chang, Ph.D.
Associate professor, Department of Pharmacology, College of Medicine, Gachon University
Director, Department of Basic Neuroscience, Neuroscience Research Institute, Gachon University
Response to the reviewer 3’s comments:
Point 1. Figure 4D legend. n=5~6 needs to be corrected to the exact number used for each group (if it meant different numbers for different experiments shown as separate graphs). Similarly, figure 5B
legend also needs the same correction. 'n' for animal number should always be accurate and precise for scientific experiments.
Answer: Thank you for your comment. According to the reviewer’s comment, we corrected to theexact number used for each group in the figure legends of Fig. 4 and Fig. 5 such as (WT-V; n=5, 5xFAD;n=5, 5xFAD-cxDHED; n=6).
Point 2. Previous review, question 11 raised a concern about how different 'n' value within experiments were accounted during statistical analysis. This does not seem to be answered. Please clarify. Answer: Thank you for your comment. We performed statistical analysis using GraphPad Prism 9.1.0
(221) software. Before using ANOVA we performed the normality test using Shapiro-Wilk test. When
the normality tests were passed (alpha=0.05), We performed using a one-way analysis of variance
(ANOVA) followed by Tukey’s multiple comparisons test. Also, we descripted line 404 about normality test. As well as, we confirmed the exactly significant p-value using Kruskal-wallis test because of unequal sample size.
Point 3. Sections 4.5, 4.6, revisions in red : what is "n= 3~6 and 4~6 per animal?" Did authors mean 'per group'? Please clarify.
Answer: Thank you for your comment. According to the reviewer’s comment, we corrected to the exact number used for each group in the method section 4.5 and 4.6 such as (n= 3–6 per group) or (n=4–6 per group).
Point 4. Revised lines 35-37 in introduction: 'Although several drugs for AD were continuously developed and expanded in clinical trial, there is no prevention or medicine for a variety reasons including side effect"- this statement is incomplete and vague. The introductory section is very important with regard to giving major context details. 'variety reasons including side effects'- needs significant correction.
Answer: We appreciate your important comments and agree with your comments. We revised the relevant part of the manuscript as “Although several different types of drugs for AD, such as nanomedicines or monoclonal antibodies, have been continuously developed and expanded in clinical
trials, these were complicated by toxicities or unanticipated side effects, and as such there is still no definite prevention or medicine [7].” and highlighted them in red in the resubmitted mansucript. Point 5. Pointing out few instances which needs grammatical corrections: line 40, lines 58-59, line 74, line 132, line 149, line 205, line 227, line 298. There are many examples. Hence, as indicated above, authors should definitely consider conducting an extensive language and grammar check and make necessary corrections.
Answer: Thank you for your comments. According to your review, we asked again professional native speaker(s) in an English editing company (“Editage”) to revise the language of the manuscript although the submitted manuscript was already checked by professional editorial institution.

Round 3
Reviewer 3 Report
The overall quality of the manuscript has increased compared to the previous versions and authors have taken good efforts to answer all the questions raised in the previous review. Questions with regard to the animal number, and statistical tests have been answered and appropriate corrections have now been made. The authors have also done an extensive language check which is indeed reflected throughout the manuscript. The manuscript in the current form can be recommended for publication in the esteemed journal, IJMS.
Author Response
We appreciate all the comments and the work of the reviewer.